# Management of Cutaneous Squamous Cell Carcinoma of the Scalp: The Role of Imaging and Therapeutic Approaches

**DOI:** 10.3390/cancers16030664

**Published:** 2024-02-04

**Authors:** Júlia Verdaguer-Faja, Agustí Toll, Aram Boada, Álvaro Guerra-Amor, Carla Ferrándiz-Pulido, Ane Jaka

**Affiliations:** 1Department of Dermatology, Hospital Universitari Germans Trias i Pujol, 08916 Badalona, Spain; jverdaguerfa.germanstrias@gencat.cat (J.V.-F.); aboada.germanstrias@gencat.cat (A.B.); 2Departament de Medicina, Universitat Autònoma de Barcelona, 08035 Barcelona, Spain; 3Department of Dermatology, Hospital Clínic de Barcelona, Universitat de Barcelona, 08036 Barcelona, Spain; atoll@clinic.cat; 4Department of Dermatology, Hospital Universitari Vall d’Hebron, 08035 Barcelona, Spain; alvaro.guerra@vallhebron.cat

**Keywords:** squamous cell carcinoma, cutaneous squamous cell carcinoma, scalp, head and neck, treatment, management, surgery, margins, recurrence, imaging

## Abstract

**Simple Summary:**

Cutaneous squamous cell carcinoma is the second most common subtype of skin cancer. The scalp is one of the most frequently affected locations and is associated with a higher risk of complications, compared to other locations. In addition, it has a characteristic thickness and anatomical structure that may influence both growth pattern and treatment of primary cutaneous squamous cell carcinoma; while clinical peripheral margins may be easily achieved during surgery, vertical excision of the tumor is limited by the skull. Despite having a unique anatomy, current guidelines do not offer specific recommendations for cutaneous squamous cell carcinoma of the scalp, which may lead to inconsistent decision-making in multidisciplinary committees when discussing tumors with some risk factors or with close histological margins. Thus, more data are needed to improve its management and assist multidisciplinary teams in shared decisions.

**Abstract:**

Cutaneous squamous cell carcinoma (cSCC) is the second most common subtype of skin cancer. The scalp is one of the most frequently affected locations and is associated with a higher rate of complications, compared to other locations. In addition, it has a characteristic thickness and anatomical structure that may influence both growth pattern and treatment of primary cSCC; while clinical peripheral margins may be easily achieved during the surgery, vertical excision of the tumor is limited by the skull. Despite having a unique anatomy, current guidelines do not contemplate specific recommendations for scalp cSCC, which leads to inconsistent decision-making in multidisciplinary committees when discussing tumors with high risk factors or with close margins. This article provides specific recommendations for the management of patients with scalp cSCC, based on current evidence, as well as those aspects in which evidence is lacking, pointing out possible future lines of research. Topics addressed include epidemiology, clinical presentation and diagnosis, imaging techniques, surgical and radiation treatments, systemic therapy for advanced cases, and follow-up. The primary focus of this review is on management of primary cSCC of the scalp with localized disease, although where relevant, some points about recurrent cSCCs or advanced disease cases are also discussed.

## 1. Introduction

### 1.1. Cutaneous Squamous Cell Carcinoma

Cutaneous squamous cell carcinoma (cSCC) is a malignant tumor that originates from the keratinocytes of the epidermis or the hair follicles, is locally invasive, and has the potential to metastasize [1]. It is the second most common subtype of skin cancer, accounting for 20–50% of keratinocyte carcinomas [2,3,4,5], and represents the second cause of death from skin cancer, after melanoma [6]. The etiology of cSCC is multifactorial, with both environmental and host factors implicated, but it is especially related with solar ultraviolet radiation and immunosuppression [1,3,7,8,9,10]. Thus, cSCC occurs more frequently in elderly white men (ratio of men–women of 2:1), especially in chronic sun-exposed areas [2,4,5,8,9,10,11,12,13,14].

Surgery remains the treatment of choice for cSCC, with a 5-year disease-free survival rate ≥91% [3,11,13,15,16,17,18]. Although most cSCCs are successfully treated with surgery, there is a subgroup of tumors that present with cSCC-related complications [3,10]: local invasion, local and regional recurrences, and lymph node metastases, with distant metastasis being rare [18,19] and the mortality rate due to cSCC being quite low (2–3%) [7,18,20,21,22,23].

### 1.2. Staging System in Cutaneous Squamous Cell Carcinoma

Currently, two main staging systems are used to predict the risk of cSCC-related events (recurrence, nodal metastasis, and/or death): the American Joint Committee on Cancer’s 8th edition staging system (AJCC8) and the Brigham and Women’s Hospital T classification system (BWH) [2,7,9,24,25,26,27]. Although the BWH system does not address nodal and metastasis classifications or advanced stage groups, it seems to be more accurate than the AJCC staging system in the classification of localized cSCC [9,10,24,25,27,28].

cSCC is considered high-risk when staged as T3/T4 with the AJCC8 staging system. However, a limitation of this staging system is that the T4 group is less frequently used, as few tumors meet the inclusion criteria. Thus, most events occur in the T3 group, but a substantial proportion of these behave well, leading to a heterogeneous group unable to detect those T3 tumors associated with poorer outcomes. On the contrary, the BWH staging system stratifies T2 tumors into a low-risk T2a stage and a high-risk T2b stage, providing superior prognostication for patients with localized cSCC patients [9,24,25,26,27]. Karia et al. demonstrated BWH T2b/T3 tumors account for 70% of nodal metastases and 85% of disease-specific death [28]. Therefore, those cSCCs classified as T2b/T3 by BWH T are considered high-risk cSCCs [2,9,24,26,27,28]. Finally, the group of Salamanca proposed an alternative system to classify the AJCC8 T3-stage more accurately and identify subgroups of higher-risk patients (T3b and T3c) [29].

## 2. Cutaneous Squamous Cell Carcinoma of the Scalp

### 2.1. The Scalp: A Special Location

Most primary cSCCs are located on the head and neck area (approx. 35–70%) [4,5,8,9,12,30], which is associated with a higher risk of developing recurrences or metastasis compared to other locations [7,8,11].

Despite comprising less than 5% of the body surface area, the scalp accounts for 3–20% of all cSCCs [4,5,11,12,13,31]. cSCC of the scalp develops more frequently in men and elderly people [5,12]. This could be explained to the chronical sun-exposure of the scalp and the protective role of the hair, scarcer in men due to common male baldness and to cultural preferences (i.e., shorter hair for men) [5,12,32].

Some authors have supported the inclusion of the scalp as a high-risk location, as it is associated with worse prognosis [31,33,34], including a risk of local recurrence of 6–10% [32,35,36], and a 7–9% risk of lymph node metastasis [32,33].

### 2.2. Scalp Anatomy

The scalp exhibits characteristic anatomical and pathological features that makes it unique. It extends from the superior occipital line to approximately 2 cm below the frontal hairline, and has a stratified structure composed of five basic layers—skin (composed of epidermis and dermis), subcutaneous tissue (or dense connective tissue), epicranial aponeurosis or galea aponeurotica, loose areolar tissue (or loose connective tissue), and periosteum (or pericranium)—overlying the skull (Figure 1). In addition, it has a closely arranged adnexa (sebaceous glands, hair follicles, and eccrine and apocrine glands) surrounded by a dense network of blood vessels, lymphatics, and nerves that course through the subcutaneous layer. The galea aponeurotica is a firmer layer of the scalp and is continuous with the frontalis muscle anteriorly, the occipitalis muscle posteriorly, and the temporoparietal fascia laterally. The skull beneath the scalp is composed of separate cranial bones—frontal, temporal, parietal, and occipital bones—which are used to virtually divide the scalp into sections [31,34,37].

This unique anatomical structure may influence both behavior and treatment of the primary tumors of the scalp. Its different layers, especially the firmer ones—galea and periosteum—may condition the growth pattern of the tumors due to the innate resistance to infiltration they are believed to have, favoring the lateral dissemination of neoplastic cells, but preventing vertical growth [31,37]. On the other hand, while clinical peripheral margins may be easily achieved during the surgical procedure, vertical excision is limited by the skull [31,35].

Another peculiarity of the scalp is its lymphatic drainage. The frontal part of the scalp drains into the parotid, submandibular, and deep cervical lymph nodes, while the posterior portion drains into the posterior auricular and occipital lymph nodes [37]. The parotid gland is the most common drainage site for tumors in the anterior scalp [38], and the involved nodes are usually located in the superficial lobe. However, some may be located beyond the facial nerve level while others are located above the parotid fascia.

Despite the peculiarities of this location, current guidelines do not recommend a specific management for cSCCs of the scalp, leading to inconsistent decision-making in some cases.

### 2.3. Clinical Presentation and Diagnosis

Invasive cSCCs of the scalp may have different clinical presentations depending on tumor size, differentiation, and skin type, but usually appear as a rapidly growing pink-reddish hyperkeratotic plaque or tumor, with or without a central horn plug, that may ulcerate or associate crusts, or as a chronic, non-healing ulcer. It commonly arises on chronic sun-damaged skin, typically in hairless areas of the scalp of males, associated with the presence of actinic keratoses (over an area of “field cancerization”, a marker of risk, although the rate of transformation itself of individual solar keratoses is low). On dermoscopy, in situ cSCCs are characterized by yellowish/white opaque scales and clusters of small dotted and glomerular vessels. When progressing to invasive cSCC, looped/hairpin and/or polymorphous vessels over an erythematous/whitish background may be typically identified, although some glomerular/dotted vessels can still be seen.

In addition, scale/keratin and white circles are typically seen in those well-differentiated cSCCs, whereas ulceration and blood spots are more common in poorly differentiated tumors (Figure 2 and Figure 3) [2,8,9,31,39,40].

Histological confirmation is the gold standard for the diagnosis, showing an atypical epithelial cells proliferation that extends beyond the epidermis into the underlying dermis, and in some cases may invade subjacent structures as well (i.e., subcutaneous fat, fascia, muscle, etc.) (Figure 4 and Figure 5) [7,9,39]. It can be assessed either by an incisional biopsy (i.e., incision or punch), shave biopsy, or directly by performing an excisional biopsy of the entire lesion, depending on the characteristics of the lesions and the judgment of the physician [10]. Due to the characteristics of the scalp, those techniques that allow obtaining a full-thickness specimen, such as incisional or excisional biopsies rather than shave biopsies, may be preferred to better assess the real tumor’s depth and infiltration [39].

When facing a cSCC of the scalp, a thorough physical examination is mandatory, for early detection of clinical risk factors or complications (such as tumor diameter >2 cm, infiltration or adherence of the tumor to underlying structures, neurologic symptoms, satellitosis, etc.), with an emphasis on regional lymph node basins, parotid and cervical, to rule out node metastasis [7,10,39,41]. The presence of a clinically palpable regional lymph node, as well as an abnormal lymph node detected by imaging exam during the diagnostic process, should lead to a fine-needle aspiration or core biopsy of the suspicious node and to additional studies for clinical staging and preoperative evaluation assessment [7] (see Section 4. Management of Regional Node Disease and Section 6.1. The Role of Imaging in Diagnosis and Staging).

### 2.4. Clinical and Pathological Risk Factors in Scalp cSCCs

There are certain clinical characteristics and histological features of a tumor that may increase the risk of developing complications and poor prognosis, such as a diameter >2 cm, the presence of perineural invasion (PNI) of nerves >0.1 mm, a poorly differentiated histological grade, or lympho-vascular invasion [2,7,9,21,24,39,42,43].

Immunosuppression is also an important risk factor and may include human immunodeficiency virus (HIV) infection, solid organ transplant, hematopoietic stem cell transplant, or chronic lymphocytic leukemia. Several studies have shown worse outcomes for cSCCs in immunosuppressed patients, with a higher risk of locoregional recurrences, metastatic cSCC, and cSCC-related death [9,12,44,45,46,47].

All risk factors described in the literature are summarized in Table 1. These features allow stratifying cSCCs into low-risk and high-risk tumors, and identifying those cSCCs with a more aggressive behavior and a higher risk of recurrence and metastasis that may benefit from a closer follow-up and specific management.

## 3. Surgical Treatment

Surgery remains the treatment of choice for cSCCs, and mainly includes wide local excision with postoperative margin assessment and Mohs micrographic surgery (MMS) [3,7,10,39,51,52]. Generally, low-risk primary cSCCs are treated with conventional surgical excision, whereas high-risk cSCCs would be candidates for MMS [7,10,39,51], though this technique is not evenly available. Different surgical treatment modalities are described below and also summarized in Table 2.

### 3.1. Surgery of the Scalp: Some Initial Considerations

When beginning any scalp procedure, it is important to properly prepare the surgical field. Shaving the hair of the affected area up to at least 1 cm from the suture margin—if applicable—and using a towel wrap, or cutting or pinning down perilesional hair with clips or tape, may prevent the introduction of foreign bodies into the wound [34,60].

It must be considered that the scalp is supplied by a rich network of anastomosing arteries within the subcutaneous layer, which are fixed to fibrous septa and often bleed profusely during surgery (Figure 1). In this regard, allowing at least 10 min between the injection of lidocaine with epinephrine and the first incision may provide better visualization and hemostasis during the surgery, facilitating the procedure and the identification of tissue planes and vulnerable structures [34]. In addition, the use of tumescent local anesthesia may also facilitate the procedure, expanding and allowing easier plane dissection with lower risk of bleeding [34,61]. Later, during the intervention, those blood vessels affected may be manually compressed, located, and either ligated with suture or sealed with electrocautery. The anesthetic infiltration should be subcutaneous or intradermal since deeper injections below the galea do not anesthetize the scalp [60].

It should be noted that collaboration may be needed when surgically approaching large scalp tumors. Particularly, when tumors invade bone or in sentinel lymph node biopsy, cooperation between dermatologists and plastic surgeons, neurosurgeons, and/or head and neck surgeons may be essential [34].

### 3.2. Curettage and Electrodesiccation

Curettage and electrodesiccation consists of scraping away tissue with a curette down to a firm layer of normal dermis and denaturing the area by electrodessication, with up to three cycles, in three different directions, performed in a session. It is a fast and cost-effective technique used in daily practice for the treatment of low-risk cSCCs, but no randomized controlled trials nor prospective studies compare this technique with other treatments modalities. Small studies have described good responses in selected lesions that are superficial, with a diameter smaller than 2 cm, and in low-risk locations [7,10,39,51].

However, the potential follicular extension of the tumor in areas that harbor terminal hair may be associated with poorer results when using therapeutical modalities that do not assess histological margins [7,10].

### 3.3. Excision with Postoperative Margin Assessment

The most common therapeutic option for cSCCs is conventional surgery with wide local excision (including a margin of clinically normal appearing skin around the tumor and the surrounding erythema), followed by postoperative histological evaluation of margins with “bread-loaf” histopathologic sectioning technique. Despite that no randomized controlled trials comparing different excision margins for cSCCs have been performed, the current evidence is based on retrospective studies and some systematic reviews, with generally good prognosis [7,10,39,51]. From the current literature, an overall local recurrence risk of 3–16% (with most of studies reporting risks ≤6%) and a regional metastasis risk of 1–4% after conventional excision is deduced [3,11,15,17,18,20,21,23,32,62]. This nodal metastasis risk increases to 5–14% in head and neck cSCCs [2,11,15], and to 7–9% on the scalp [32,33].

Achieving clear surgical margins is the most important treatment consideration for patients with cSCCs amenable to surgery [51]. Some works, mainly retrospective and based on cSCCs removed with MMS, analyzed the subclinical extension of the tumor and the number of stages needed, in order to estimate the width of clinical margin required to achieve histologically clear margins in a standard surgery of cSCC [53,63,64]. Based on these findings, reference guidelines for cSCC management present similar recommendations about the clinical peripheral margins to be performed during conventional excision in primary cSCCs, to ensure complete removal in ≥95% of cases: between 4–10 mm depending on the risk factors (specially tumor diameter and location) [7,13,39,51,65].

The scalp has been proposed as a high-risk location [7,33,35]. As mentioned before, excision with comprehensive intraoperative margin control is the preferred surgical technique for high-risk cSCCs [7,10]. However, this technique is not available in many treating centers. Thus, many cSCCs of the scalp are commonly removed by standard surgery.

NCCN guidelines state that cSCCs in high-risk locations (including the scalp) with a diameter <1 cm, 1–1.9 cm, and ≥2 cm should be removed with clinical margins of at least 4 mm, 6 mm, and 9 mm, respectively [7]. However, larger excision margins should be considered when other risk factors are present (i.e., poor differentiation, PNI, or invasion of subcutaneous tissue). Nonetheless, the European consensus group suggests at least a minimum of 5 mm clinical safety margin for any cSCC [51], and the British Association of Dermatology (BAD) guidelines recommend ≥6 mm for high-risk, and ≥10 mm for very high-risk cSCCs [39]. Thus, in our opinion, any cSCC of the scalp should be removed with a minimum peripheral margin of 5–6 mm, extending to ≥10 mm for those cases with other high-risk factors.

Another important issue when talking about standard excision of cSCC of the scalp is the deep surgical plane that should be reached during the procedure. There is no consensus in current cSCCs guidelines, and different planes have been proposed, without enough evidence to support them: the hypodermis (assuming that the deeper layers are not macroscopically affected), the inclusion of a thin layer of subcutaneous fat or reaching “the next clean surgical plane” [7,10,39,51,52,63,65].

The BAD, European, and Scottish guidelines recommend including the galea aponeurotica in the excision, due to its firm consistency and the probable innate resistance to infiltration [39,51,52]. Moreover, deeper surgical planes beyond the galea are associated with lower rates of close/affected margins [35].

Another consideration to keep in mind during scalp cSCC surgery is that the majority of the scalp mobility relies on the loose areolar tissue layer, while most of the nerves and vasculature lay above it. Thus, this layer can be easily dissected and would also be a relatively safe plane during dissection [37].

Considering all the above, it seems wise to recommend excision below the galea aponeurotica, both to reduce margin positivity and the risk of complications.

Finally, taking a deeper margin should be considered if there is clinical concern of an incomplete resection during the surgery [39]. When suspicion or evidence of tumor invasion of bone—clinically seen as subtle pitting of the bone or suggested by imaging studies (see Section 6. Imaging Approach)—the outer table of the skull should also be removed [34,39].

### 3.4. Histological Margins

Little is known regarding histological margins after conventional excision in cSCC [36,66,67]. Although recommendations for incomplete excisions seem to be clearer, there are no guides for the management of those cSCCs completely removed, but with close histological margins. This is especially relevant on the scalp, where deep removal of the primary tumor is limited by the skull. In fact, three retrospective studies which analyzed cSCCs removed by standard surgery found that 4–11.9% of cases were not completely excised. One of the most common locations for these cases was the scalp (16–38%), and mostly related to the deep margin [11,68,69,70].

Globally, two main scenarios, requiring different approaches, can be faced when evaluating histological margins after the excision of a cSCC, regardless the location, by conventional surgery.

*Peripheral or deep positive margins*. Local and regional recurrences, as well as pathological positivity after re-excisions, are higher in these group of patients [36,68,71]. Thus, most guidelines recommend, when possible, re-excision as the treatment option of choice, commonly yielding clean margins [16,35,51]. If available, MMS should be the treatment of choice, rather than re-excision with postoperative margin assessment, to ensure free histological margins and avoid complications, especially in tumors with high-risk factors [34,39]. When surgery is not feasible, other treatments, such as radiotherapy (RT), might be considered [34,39].*Free but close histological margins* (by consensus, those margins between 0.1–0.9 mm, according to the Royal College of Pathologists and the BAD [39,72]). While there is scarce evidence in the literature regarding the conduct in this scenario [7,10,39,51,52,65], the British and the Scottish guidelines recommend discussing those cSCCs with histological margin <1 mm in a multidisciplinary tumor board to assess the need or not of further adjuvant treatments [39,52,72]. Thus, they consider observation in those pT1 cSCC with <1 mm histological margins in immunocompetent patients [39].

Regarding the scalp, only one retrospective study, by Jenkins et al., compared the differences in local and regional recurrence rates of cSCCs with clear but close deep margin (0.1–1.9 mm) to cSCCs with a thicker deep margin (2–6 mm and >6 mm). They observed a greater number of local recurrences in the first group (8% vs. an overall rate of 3%) [32].

Although the current evidence is scarce, careful consideration should be given to those cSCCs of the scalp with clear but close peripheral or deep margins, and re-excision or further treatments should be considered if other high-risk features are present [39]. Nevertheless, further studies assessing the role of close histological margins in relapse remain indispensable.

### 3.5. Mohs Micrographic Surgery or Excision with Complete Circumferential Peripheral and Deep Margin Assessment

MMS is a technique that has proven to be effective for the removal of skin tumors located in compromised areas in which saving tissue is essential and/or when we want to ensure negative margins in a tumor [51,73]. It is also a technique of choice for tumor subtypes that are associated with an increased risk of recurrence. In this sense, cSCCs on the scalp fulfill these criteria [51].

The technique usually involves the study of frozen tissue sections. The histopathological study is carried out in tangential sections that determine the assessment of 100% of the tumor margins compared to the conventional vertical “bread-loaf” sections in which large areas of “blind” margins may remain unexplored under the microscope [73].

MMS for cSCCs has some peculiarities. These are tumors that may be quite large and difficult to process in frozen sections. On the other hand, the tumor depth is a relevant prognostic feature in cSCCs that may be difficult to register due to technical singularities of the MMS (a debulking is separated from the true margins) [7]. Furthermore, undifferentiated and small-nest infiltrative tumors may be difficult to detect in frozen sections. In this sense, some authors prefer a modified (slow, 3D histology) MMS technique with paraffin sections that also allow routine immunohistochemical stainings [51].

MMS has been demonstrated to be an effective technique for the treatment of high-risk cSCCs in large cohort studies, both retrospective and prospective [51,53,74,75,76,77,78,79], and is recommended by the European and American guidelines in these subsets of patients [7,51,80].

In a large multicenter prospective case series study with 1263 patients treated with MMS, in which almost all tumors were located in the head and neck area (96.5%), a risk of recurrence of 3.9% was observed after a 5-year follow-up period. The risk of recurrence was lower in patients with primary cSCC (2.6%) than in those with recurrent tumors (5.9%) [53]. Another prospective cohort study by Tschetter et al. including 745 tumors showed 5-year local recurrence—free survival, nodal metastasis-free survival, and disease-specific survival of 99.3%, 99.2%, and 99.4%, respectively [76]. Finally, a recent Spanish prospective study including 371 cSCCs reported recurrence rates of 4.5 per 100 person-years [79].

As the recurrence following MMS is low, the risk can be increased by several factors, including unfinished surgery, the number of stages needed, immunosuppression [79], invasion beyond the subcutaneous fat, poor histological differentiation [74], and PNI [81].

Although there are no randomized prospective clinical trials comparing MMS with conventional surgery, several retrospective studies have shown significantly lower recurrence rates for MMS [51,54,55]. Interestingly, in the largest comparative retrospective study by Van Lee et al., which included 672 head and neck cSCCs (approximately 20% on the scalp), the overall recurrence rate was 8% after standard excision vs. 3% for MMS [54]. Recent studies have also demonstrated that MMS is more cost effective than wide local excision and would be particularly indicated for high-risk cSCCs [56].

Studies reporting MMS for scalp cSCCs have reported cure rates and recurrence rates that are equivalent to those reported in other areas [57,58]. The first stage of MMS of scalp cSCCs should include the subcutaneous tissue and run into the subgaleal plane [59].

### 3.6. Reconstructive Approaches on the Scalp

The preferred options in the scalp, as a high-risk location, are those closures that do not rotate tissue around and/or alter anatomy of the surgical bed, where “residual cells” of the tumor could remain. Primary closure (linear repair), skin grafting (split- or full-thickness), the use of dermal matrices, and secondary intention healing with granulation are appropriate reconstructive approaches, especially if MMS is not available [10,34]. Careful consideration might be given to skin grafting when the periosteum is removed and the bone is exposed, or if previous RT has been performed over the area [34,37,82].

Nonetheless, reconstruction of medium-sized (2–5 cm width) and larger defects on the scalp after cSCC extirpation can sometimes be challenging due to tightness of the surrounding soft tissues and a lack of soft tissue reservoir [34]. In those cases, with a large defect or in which the periosteum or the outer table of the skull has been removed, tissue rearrangement with flaps might be required. These large closures should be delayed until negative histologic margins are confirmed [7,10,39,53].

Particularly in the scalp, subcutaneous stiches are not usually used in order to avoid damage of the hair follicles and prevent local alopecia. On the contrary, the use of staples might be of interest as it is faster to apply, more hygienic, and also allows daily washing, essential to reducing local inflammation and infection [60].

## 4. Management of Locoregional Disease

### 4.1. Management of Patients with Satellitosis or In-Transit Metastases

Satellitosis or in-transit metastases (S-ITM) are nonepidermal lesions originating between the primary tumor and the first tumor-draining lymph nodes, considering as satellitosis those that occur within 2 cm of the primary tumor. They are thought to occur as a consequence of intralymphatic or possibly angiotropic tumor spread [83], and are more common in immunosuppressed patients [83,84]. S-ITM have proved to be an independent poor-outcome risk factor in cSCCs [83,85,86,87], with outcomes comparable to node-positivity in terms of recurrence and disease-specific survival [86]. A recent study of Marti-Marti et al., demonstrated that the size (≥20 mm) and the number of lesions (>5) of S-ITM are two independent prognostic factors for relapse, and the number of lesions for specific death [83].

The head and neck region is the most common location of S-ITM [83,84], occurring more commonly in the scalp (Figure 6) [84].

Although uncommon, S-ITM represent an authentic management challenge as they are not included in current cSCC staging systems and guidelines [7,39,51]. According to the literature, S-ITMs are usually excised by surgery with or without adjuvant RT, and less frequently with RT as monotherapy or systemic therapy [83,84]. Surgery followed by adjuvant RT seems to obtain better outcomes [83], although studies comparing treatment approaches are lacking. If resection is not possible, systemic therapy and/or RT, if feasible, could be considered.

### 4.2. Management of Patients with Clinically Detected Lymph Nodes

Lymph node dissection is the standard of care for patients with regional lymph node metastases detected on physical examination or following imaging tests. The extent of lymph node dissection is a controversial issue. There is a growing trend to offer more conservative surgeries that provide less morbidity and better functional results without leading to worse survival [88,89]. In case of parotid lymph node involvement, superficial parotidectomy and ipsilateral cervical dissection are usually recommended since studies have proved that involvement of the parotid gland correlates with higher incidence of occult metastases in the neck lymph nodes [90]. However, these decisions must be made in the context of a multidisciplinary tumor board, considering tumor aggressiveness, patient’s status, and surgical conditions.

Adjuvant RT following lymph node dissection should be recommended if extracapsular extension is observed, more than two nodes are affected, or one node is larger than 3 cm (AJCC 8th N2 or N3) [91,92].

In patients with incompletely excised lymph node metastases (LNM) or those who are inoperable, RT and/or systemic treatment should be considered. However, the management of macroscopic lymph node disease could be soon redefined if the use of neoadjuvant treatment is expanded [93,94].

### 4.3. Management of Patients without Clinically Detected Lymph Nodes

In patients without evidence of lymph node dissemination, there is no evidence of the usefulness of elective lymphadenectomy, so this approach is not recommended.

Sentinel lymph node biopsy (SLNB) is used in various skin cancers for the early detection of LNM before they could be clinically detected. In the case of cSCC, these occur in 8–12.3% of cases, according to the systematic review by Tejera [95] and the meta-analysis of Schmitt [96]. Both groups proved that the risk of sentinel node involvement increases with the tumor stage [97]. Therefore, risk factors for SLNB positivity include tumor diameter, tumor thickness, lymphovascular invasion, PNI, or the simultaneous presence of multiple risk factors. According to Tejera and Schmitt, the false negative rate of SLNB in cSCC is 3.9–4.6% [95,96].

The role of SLNB in high-risk cSCC is currently under debate. Evidence of its usefulness comes from several studies and systematic reviews, but there are no clinical trials that have assessed its value. cSCC is a cancer prone to an orderly dissemination. Most patients showing poor prognosis develop LNM. Thus, identifying those patients with microscopic nodal disease may impact in its management.

Few studies have analyzed the association between SLNB status and survival, although it seems that there would be worse survival in patients with positive SLNB [98,99]. It is not clear if SLNB followed by completion lymph node dissection improves survival [98]. Studies that have evaluated the ideal time to perform SLNB have not shown differences in the rate of SLNB detection between performing SLNB at the same time of tumor excision or delaying it [100].

Nowadays, clinical practice guidelines do not recommend SLNB to be used routinely in cSCC. However, its adoption is encouraged in the setting of clinical trials in high-risk patients.

## 5. Non-Surgical Treatment

### 5.1. Radiation Therapy

Radiotherapy (RT) is a useful therapeutic tool to treat scalp cSCC. Several modalities/devices are used, such as external RT and brachytherapy. The relatively flat surface of the scalp makes the use of external beam RT suitable for large lesions in this area. RT can be used as primary (radical) therapy, in adjuvancy, or for palliation. Although no prospective randomized controlled trials comparing therapeutic modalities have been performed, surgery is usually recommended to treat most cSCCs. Radical RT is usually restricted to patients who reject surgery, those with severe co-morbidities or frailty, or patients with unresectable tumors. The response rates of RT are high, especially for small, superficial lesions in immunocompetent patients. Moreover, the functional and cosmetic results of RT are usually excellent. However, the 5-year local recurrence rates are above 6% and 10% for primary and recurrent tumors, respectively [15,101]. Poorer local control rates and higher recurrences are observed in higher T-stages, with T4 cSCC of the head and neck 5-year local control rates of 50–60% [15,102]. When bone is involved, the initial local control descends to 40% [103]. However, RT is a convenient option in elderly frail patients with bone invasion in which a complex neurosurgery would be necessary for radical control of the disease. In this sense, dramatic responses of giant cSCC of the scalp with extensive bone destruction have been reported [104].

Re-excision should be encouraged in patients with cSCC with positive residual microscopic margins (R1). If further surgery is contraindicated, RT is associated with lower recurrence rates than a wait-and-see conduct [105].

Adjuvant RT after complete surgical resection of scalp cSCC (R0) can be useful in some patients. Similar to other head and neck cSCCs, RT is recommended after elective node dissection of patients with metastatic neck regional dissemination, especially in those with multiple nodal metastases, nodes larger than 3 cm, or nodes smaller than 3 cm but showing extracapsular extension (N2). Adjuvant RT after parotidectomy is also indicated in patients with metastatic intraparotideal lymph nodes. Prophylactic cervical lymph node dissection or cervical RT are recommended in patients with intraparotideal metastases, and both treatments have shown similar outcomes [106].

Adjuvant RT to the tumor bed can also be considered in patients with large or named nerve involvement and in those with microscopic extensive or larger than 0.1 mm nerve invasion [107,108].

The utility of RT in adjuvancy in patients with negative surgical margins but poor prognostic features is controversial [109,110]. A recent study by Ruiz et al. has shown that RT reduces the risk of local and locoregional recurrences to half in tumors of high T-stage cSCCs (BWH T2b or T3) [111]. The American Society for Radiation Oncology and the Head and Neck Cancer International Group (HNCIG) have recently published guidelines for definitive and postoperative RT for cSCC [112,113].

Some final special considerations have to be made for RT of scalp cSCC. RT induces alopecia in hair-bearing individuals. Moreover, RT may infrequently induce bone exposure in the scalp in elderly patients with skin atrophy.

### 5.2. Systemic Treatment

#### 5.2.1. Immunotherapy with Checkpoint Inhibitors

##### 
Treatment of Advanced cSCC


cSCC harbor a high mutational burden due to UV that make them good targets for immunotherapy [114]. Recent published phase II trials support the efficacy and safety of immunotherapy (cemiplimab and pembrolizumab) in patients with locally advanced cSCC and metastatic cSCC [51,115,116,117].

Cemiplimab, an anti-PD-1 antibody, was approved by the Food and Drug Administration (FDA) in 2018 and by the European Medicines Agency (EMA) in 2019 for metastatic and locally advanced cSCCs that are not amenable to curative surgery or radiation therapy [118]. Pembrolizumab, another anti-PD-1 antibody, has also been approved for metastatic and locally advanced cSCC by the FDA in 2020, but not by the EMA.

##### 
Neoadjuvant Therapy


A recent Phase 2 multicenter study with resectable stage II-IV cSCCs in 79 patients explored the utility of neoadjuvant cemiplimab (350 mg every 3 weeks for up to four doses) before a curative intent surgery. In 40 patients (51%), a complete pathologic response was observed [93]. However, further studies with larger samples are needed. NCCN guidelines are actually considering neoadjuvant therapy after multidisciplinary discussion for selected cases [7].

#### 5.2.2. EGFR Inhibitors

EGFR inhibitors block the intracellular MAPK pathway. Of all available targeted EGFR inhibitors, those monoclonal antibodies that target the extracellular domain have been mainly used for advanced cSCC (cetuximab and, less frequently, panitumumab) [119,120,121]. Nevertheless, no clinical evidence has been demonstrated, and they are not currently approved by either the FDA or the EMA.

#### 5.2.3. Conventional Chemotherapy

The most commonly used regimens classically for cSCC are 5-fluorouracil (5-FU)/cisplatin, 5-FU/carboplatin, or paclitaxel/carboplatin combinations. Observational studies show <80% remission for combination treatments and <60% remission with monotherapy.

However, sustained remissions are rare and traditional chemotherapy is poorly tolerated by frail elderly patients who comprise the majority of those with advanced cSCC [122].

## 6. Imaging Approach

### 6.1. The Role of Imaging in Diagnosis and Staging

There is currently limited evidence regarding the need to perform imaging tests in cSCC, both in terms of indication and the most appropriate technique, and, to the best of our knowledge, there are no studies specifically evaluating the performance of imaging in the scalp.

The risk of LNM in cSCC is relatively low [21] and indiscriminate imaging could lead to a considerable number of false positives and unnecessary additional procedures [123,124]. However, it is well established that tumors with higher T scores in staging systems have a higher risk of LNM [28], and there are studies suggesting that early detection of LNM when fewer lymph nodes are affected [125], or where nodes are smaller and there is no extracapsular invasion [126], may lead to a better prognosis. Some studies have shown a trend towards larger lymph nodes in patients with clinically detected LNM compared to those routinely screened with ultrasound [127].

Both the latest European and NCCN clinical practice guidelines agree that a complete anamnesis and detailed physical examination at the time of diagnosis should be performed in all patients, which may be sufficient in patients with low-risk cSCC [7,9].

There is evidence, nevertheless, that patients with high-risk cSCC with normal physical examination may also benefit from imaging to detect subclinical metastatic disease. In three retrospective cohort studies at Brigham and Women’s Hospital using mainly computed tomography (CT) scans in patients with high-risk cSCC (≥T2b-BWH), imaging was associated with a change in therapeutic approach in up to one-third of patients [128,129,130], and it was associated with a decrease in the number of poor outcomes [128]. Furthermore, another retrospective cohort study demonstrated a higher sensitivity of ultrasound than physical examination alone for the detection of lymph node metastases, although at the expense of a higher false positive rate [124].

In the light of this evidence, European guidelines recommend imaging tests at diagnosis in patients without palpable lymphadenopathy on examination who present with high-risk cSCC, defined by a T score according to BWH equal to or greater than T2b or the presence of any of the risk factors proposed by the EADO [9].

Regarding the type of technique for nodal staging, guidelines recommend preferably ultrasound or contrast-enhanced CT [9]. There are no specific recommendations for high-risk cSCC of the scalp, but in our view CT with contrast is probably a more efficient technique than ultrasound, as it allows assessment of not only lymph node involvement but also the depth of the tumor, which in some cases may be greater than clinically expected, as observed in studies in which this technique was performed perioperatively [130]. In tumors with features that raise suspicion of bony invasion of the calvarium (firmly adherent or pain on palpation of the bony margin), it is indispensable. Magnetic resonance imaging (MRI), although generally less available, is superior to CT in assessing deep invasion beyond the outer table of the skull, PNI, and parotid or central nervous system involvement [131,132,133]. However, a substantial portion of the population affected by high-risk cSCC of the scalp comprises elderly individuals, frequently presenting with concomitant renal failure, or kidney transplanted patients. These conditions often constrain the use of contrast-enhanced CT, thereby markedly diminishing the sensitivity of the imaging technique. In such instances, an alternative approach to evaluate lymph node involvement, such as ultrasound, may prove to be more advantageous.

Finally, in patients with lymph node involvement, imaging tests should also be performed to rule out distant metastases, such as body CT or positron emission tomography [9].

### 6.2. Follow-Up

The use of imaging tests for the follow-up of patients with cSCC is recommended in three scenarios: high-risk cSCC, locally advanced disease, and metastatic disease. In contrast, for patients with low-risk cSCC, the guidelines propose annual clinical follow-up, at least for the first 2 years, as this is the period with the highest risk of local and distant recurrence [7,9,134].

In high-risk tumors, European clinical practice guidelines propose to preferably perform lymph node ultrasound, which should include cervical and parotid lymph node territories every 3–6 months for the first two years [9]. From our point of view, ultrasound would be a good option due to its excellent cost-effectiveness and the absence of exposure to ionizing radiation for the follow-up of those cases of high-risk cSCC of the scalp in which it is not necessary to assess the status of the calvarium, i.e., those patients with completely excised tumors and no signs of local recurrence. As we have emphasized previously, the non-utilization of contrast in a population frequently burdened with associated renal insufficiency is another advantage that supports the use of ultrasound in monitoring instead of contrast CT scan.

Despite these recommendations, it is essential to consider that some studies have demonstrated that with a screening ultrasound investigation at baseline, only a few LNM are detected, while the majority of metastases are identified through clinical examination, typically self-examination, during follow-up [127]. This limitation reduces the utility of the technique and raises questions about the appropriate timing of routine ultrasound. Therefore, prospective studies assessing the true benefit of routine radiological examinations compared to thorough self-examination or clinical follow-up remain indispensable.

In patients with locally advanced and metastatic disease, a frequency of 3 to 6 months is also proposed, and the choice of imaging technique is left to the discretion of the clinician [9]. In this type of patient, either because of the need to assess the status of the calvarium or other organs at the same time, we consider it more convenient to perform a CT.

## 7. Algorithms for the Management and Treatment of Primary Scalp cSCC with Localized Disease

### 7.1. Proposed Algorithm for the Initial Management of Primary cSCC of the Scalp

Based on current evidence, an algorithm for the initial management of patients with primary cSCC of the scalp has been proposed (Figure 7). Comprehensive treatment approach of scalp cSCC should include both correct surgical excision and appropriate closure (Figure 8).

### 7.2. Proposed Algorithm for the Management of Histological Margins and Other Histological Features

Based on current evidence, an algorithm for the management of scalp cSCC according to histological features, and specially to histological margins, has been proposed (Figure 9).

### 7.3. Proposed Algorithm for the Follow-Up of cSCC of the Scalp

Based on current evidence, an algorithm for the proper follow-up of patients with cSCC of the scalp has been proposed (Figure 10).

## 8. Conclusions and Future Directions

With increasingly longer life expectancies, the health burden associated with cSCC is likely to rise still further. Although the understanding of cSCC has grown in recent years, much research remains to be conducted. The scalp has a characteristic thickness and anatomical structure that may influence both behavior and treatment of the cSCC, making a specific management mandatory. Current guidelines do not contemplate specific recommendations for cSCC of the scalp, and more data are needed to improve its management and elucidate other risk factors that might better predict prognosis and assist in shared decisions of multidisciplinary teams.

Future research directions in cSCC of the scalp should include the evaluation of the role of histological margins in recurrences, compare different approaches (surgery, RT, immunotherapy, others) in involved histological margins, study other clinicopathological or molecular factors that might predict poor outcomes, evaluate the role of sentinel lymph node biopsy in the staging of very high-risk cSCC, assess the true benefit of routine radiological examinations and best imaging technique, and compare adjuvant RT after surgical excision of high-risk cSCC.

## Figures and Tables

**Figure 1 cancers-16-00664-f001:**
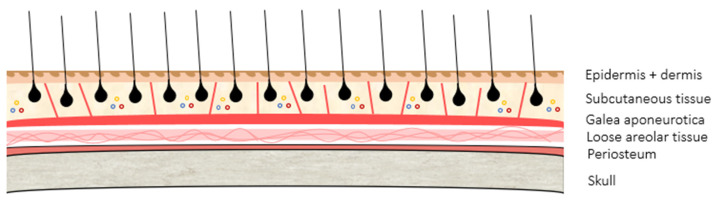
Graphical representation of the anatomical structure of the scalp, with its five layers: epidermis + dermis, subcutaneous tissue, galea aponeurotica, loose areolar tissue, and periosteum. Blood vessels, lymphatics and nerves exist through the subcutaneous layer (small color circles), adjacent to fibrous tracts.

**Figure 2 cancers-16-00664-f002:**
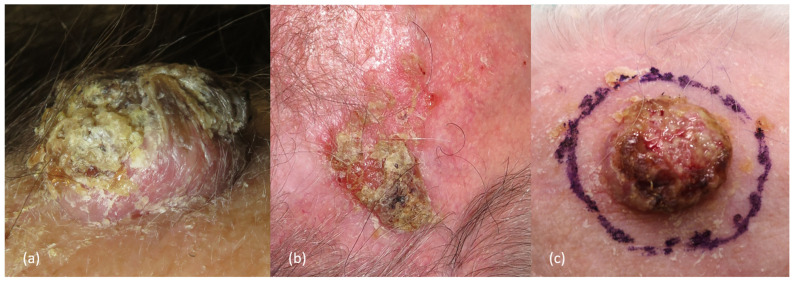
Clinical appearance of different cutaneous squamous cell carcinoma of the scalp. (**a**) Well-differentiated scalp cSCC. A rounded pink and hyperkeratotic tumor, with well-defined borders, in the parietal region of the scalp. (**b**) Moderately differentiated scalp cSCC. Hyperkeratotic erythematous plaque in the right parietal region, with poorly defined borders, and small areas with ulceration. Notice the actinic damage surrounding the lesion. (**c**) Poorly differentiated scalp cSCC. Erythematous and fleshy tumor, with a diameter greater than 2 cm, in the frontal region of an elderly patient.

**Figure 3 cancers-16-00664-f003:**
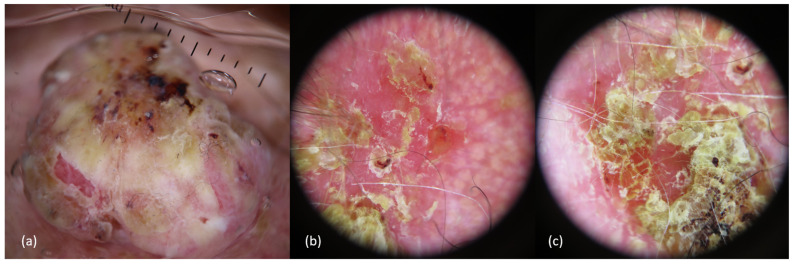
Dermoscopy of different cutaneous squamous cell carcinomas. (**a**) Keratotic tumor, with yellowish-whitish keratosis in the center, with some hemorrhagic area, and a pink peripheral rim with hairpin and looped vessels. (**b**,**c**) Hyperkeratotic lesions with poorly defined edges, with an erythematous background with yellowish scales and keratosis, and small erosions. Few dotted vessels can be seen in the center of image (**b**).

**Figure 4 cancers-16-00664-f004:**
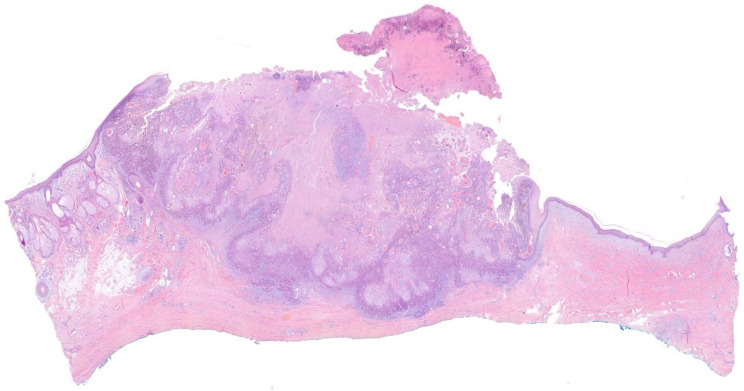
Histological image of a cutaneous squamous cell carcinoma of the scalp, showing a proliferation of atypical squamous cells, ulcerated, that infiltrate the dermis, subcutaneous tissue, and galea aponeurotica (H&E, 0.6×).

**Figure 5 cancers-16-00664-f005:**
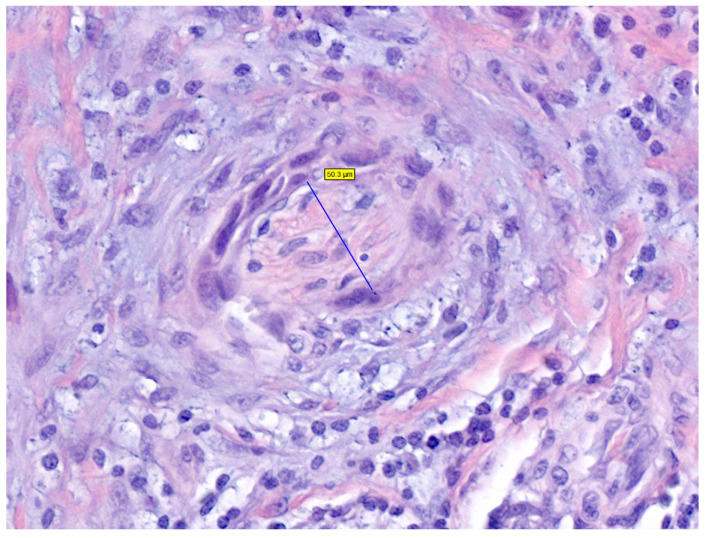
Perineural invasion of a nerve of 0.05 mm in a cutaneous squamous cell carcinoma of the scalp (H&E, 63×).

**Figure 6 cancers-16-00664-f006:**
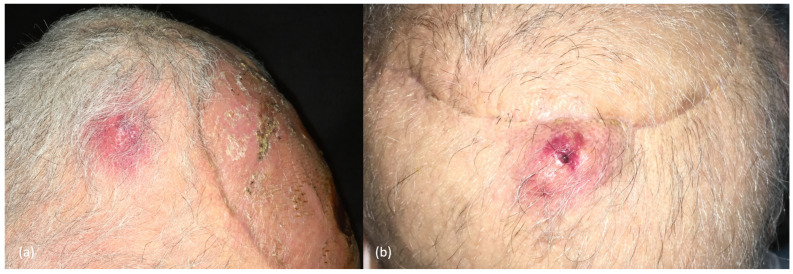
(**a**,**b**) Clinical images of two cSCC satellitosis in the scalp, in an immunosuppressed patient, appearing as erythematous nodules next to the scar of a previously excised high-risk cSCC.

**Figure 7 cancers-16-00664-f007:**
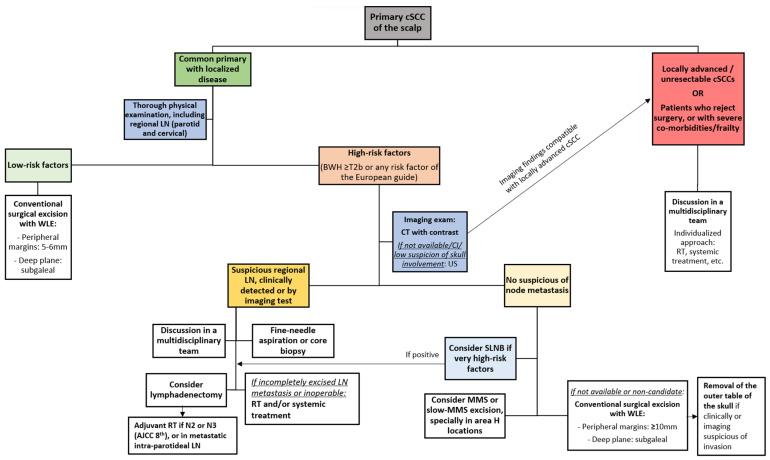
Proposed treatment algorithm for patients with primary cSCC of the scalp. cSCC: cutaneous squamous cell carcinoma; LN: lymph node; WLE: wide local excision; CT: computed tomography; CI: contraindication; US: ultrasound; RT: radiotherapy; SLNB: sentinel lymph node biopsy; MMS: Mohs micrographic surgery.

**Figure 8 cancers-16-00664-f008:**
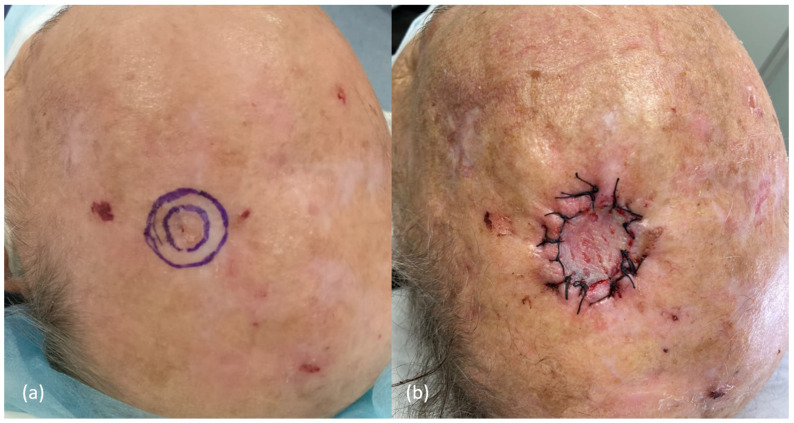
Example of the surgical approach performed in a cutaneous squamous cell carcinoma, <2 cm of diameter, in the left parietal region of an immunocompetent patient, using conventional surgery. (**a**) Peripheral clinical margins of 5 mm were performed. (**b**) As the histological margins were evaluated postoperatively, closure with a partial skin graft was chosen.

**Figure 9 cancers-16-00664-f009:**
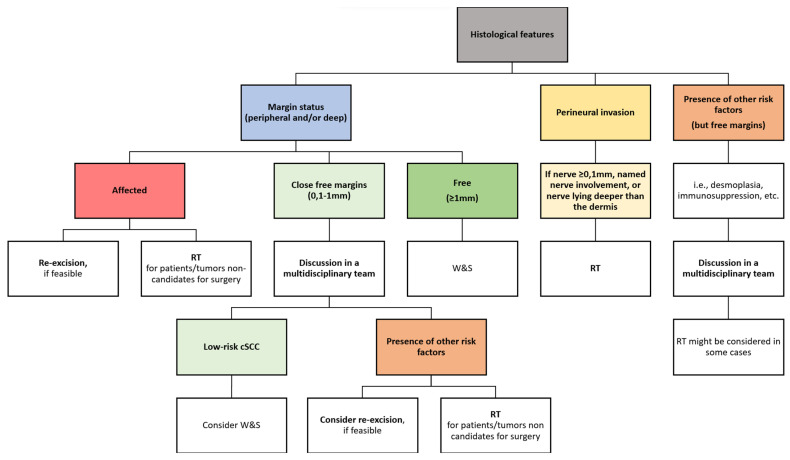
Proposed treatment algorithm for the management of histological margins and other histological features. RT: radiotherapy; cSCC: cutaneous squamous cell carcinoma; W&S: wait-and-see/close surveillance.

**Figure 10 cancers-16-00664-f010:**
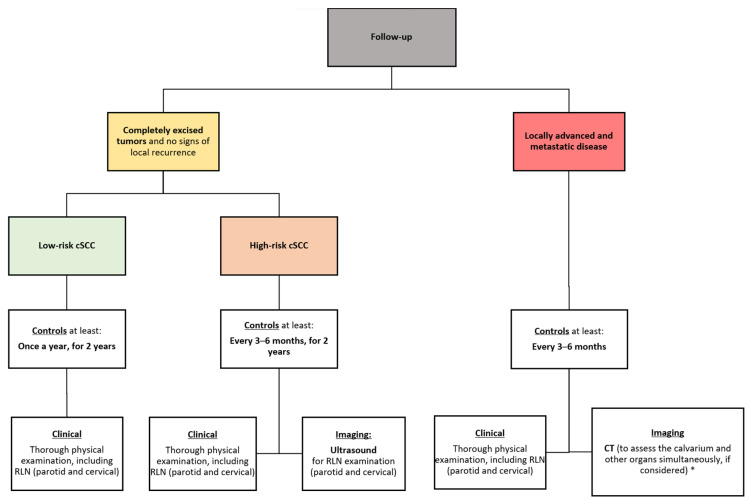
Proposed algorithm for the follow-up of cSCC of the scalp. cSCC: cutaneous squamous cell carcinoma; RLN: regional lymph node; CT: computed tomography. * However, the choice of imaging technique is left to the discretion of the clinician.

**Table 1 cancers-16-00664-t001:** Clinical and histological risk factors in scalp cSCCs.

Factors	Low-Risk	High-Risk	References
**Clinical**			[2,7,9,14,15,18,21,22,23,39]
Immune status	Immunocompetent	Immunosuppressed	
Primary vs. recurrent	Primary	Recurrent, metastatic	
Site of prior radiation therapy	No	Yes	
Site of chronic inflammation	No	Yes	
Rate of growth	Slow	Rapid	
Tumor dimensions (including peripheral rim of erythema)	Size/diameter: <2 cm	Size/diameter: >2 cm. ○ Very high if ≥4 cm.	
Tumor circumscription	Well-defined borders	Poorly defined borders	
Neurologic symptoms	Absent	Present	
**Pathologic**			[2,7,9,14,15,16,21,22,23,24,32,39,42,43,48,49,50]
Tumor dimensions	Size/diameter: <2 cm	Size/diameter: >2 cm	
Histologic grade	Well or moderately differentiated (G1-2)	Poorly differentiated (G3)	
Histologic type/Growth pattern	Subtype not otherwise specified	Acantholytic (adenoid), adenosquamous (or mucin-producing), desmoplastic, spindled, metaplastic/sarcomatoid	
Perineural invasion	Absent	Present, diameter of involved nerve ≥0.1 mm, multifocality, involvement of deep dermal nerves, or named nerves	
Lymphovascular invasion	Absent’	Present	
Tumor depth	Depth of invasion (DOI) <2 mm Anatomic (Clark) level I–III	DOI: 2–6 mm, and ≥6 mm in very high risk Anatomic (Clark) level IV–V Extension beyond subcutaneous fat	
Extension into osseus structures	Absent	Present	
Lymph node metastasis	Absent	Present, size of metastasis >3.0 cm, presence of extranidal extension, involvement of contralateral lymph nodes	
Positive margins	Absent	Present	
Tumor budding *	Grade 1: 0 to 4 buds	Grade 2: 5 to 9 buds, Grade 3: ≥10 buds	
Deep histological margin **	2–5.9 mm ≥6 mm	0.1–1.9 mm	

G: differentiation grade (from G1 well-differentiated tumors to G3 poorly differentiated ones). DOI: Depth of invasion (as measured from the granular layer of the adjacent normal epidermis to the base of the tumor). * A distinct pattern of invasion assessed by the number of isolated cancer cells or of small clusters, measured at a hot spot, including peritumoral (invasive front) and intratumoral (center of tumor) areas. Tumor bud was defined as an isolated cancer cell or a cluster comprising < 5 cells. ** Little evidence in scalp cSCC, only one retrospective observational study of Jenkins et al. [32]. Adapted from Nagarajan et al., Schmults et al., and Keohane et al. [3,7,39].

**Table 2 cancers-16-00664-t002:** Surgical treatment in scalp cSCC.

Surgical Modalities	Recurrence Rates	Surgical Margins Recommendations		References
**Curettage and electrodesiccation**		Not recommended.		[7,10]
**Wide local excision**	Local recurrence rate: ○ 6–16%. Nodal metastasis: ○ Head and neck area (including scalp): 5–14%. ○ Scalp: 7–9%.	Peripheral surgical margins	- NCCN guidelines: *depending on the tumor’s diameter (Ø):* Ø < 1 cm: ≥4 mm. Ø 1–1.9 cm: ≥6 mm. Ø ≥2 cm: ≥9 mm. - European guideline: Minimum ≥5 mm. - BAD guideline: ≥6 mm. ≥10 mm if presence of other risk factors.	[7,32,33,35,39,51]
		Deep surgical plane recommended	- European, BAD and Scottish Guidelines, and Brewer et al.: Subgaleal. - No specific recommendations in other guidelines.	[35,39,51,52]
**Mohs Micrographic Surgery**	Recurrence rate in Head and neck area (including scalp): 3–3.9%. Recurrence rate in Scalp: 3.2%.	First stage of MMS should include the subcutaneous tissue and run into the subgaleal plane.		[53,54,55,56,57,58,59]

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
