# Peer review of "Management of Cutaneous Squamous Cell Carcinoma of the Scalp: The Role of Imaging and Therapeutic Approaches"

_cancers, 2024, doi:10.3390/cancers16030664_

Round 1
Reviewer 1 Report
Comments and Suggestions for Authors
I think this is a very good review article on squamous skin cancer and specifically scalp cancer. In addition to the exhaustive review of the evidence, algorithms are presented that can help the reader and the professional in making decisions.
I believe few changes must be made and there are, in my opinion, no conceptual or methodological errors.
I just want to make a couple of comments, to improve the quality if it is relevant.
First, in line 65 it is written and referenced that "cSCC incidence is increasing worldwide…”
It is known that non-melanoma skin cancer (squamous cell and basal cell carcinoma) is a type of neoplasia that is generally under-reported and under-registered. Therefore, in many population-based epidemiological studies it is specified that these tumors are excluded from the global analysis. Also because of this, there is little literature in this issue.
Thus, any study that indicates an increase in incidence must consider that this is not due to a real increase in etiological factors but to an improvement in the registration of these cases. It's an important concept to keep in mind.
Second, I do not agree very much with the form or distribution of point 5.2, line 591. I understand the authors want to focus on the most modern systemic treatment, but both ICI and TKIs are still chemical therapy and therefore chemotherapy.
I consider it would be more useful in a review to treat the concept of systemic therapy in general, with a first section for immunotherapy, where to include both advanced and neoadjuvant treatment, a second section for EGFR inhibitors, as the main target therapy used in this type of neoplasia , and finally (and why not), a third section briefly considering what conventional (or anti-cell cycle) chemotherapy provides, as is still an option for some patients.
Author Response
Dear colleague, thank you for your suggestions and comments.
- First, in line 65 it is written and referenced that "cSCC incidence is increasing worldwide…”. Any study that indicates an increase in incidence must consider that this is not due to a real increase in etiological factors but to an improvement in the registration of these cases. It's an important concept to keep in mind.
- Response: The sentence “cSCC incidence is increasing worldwide…” has been removed to avoid misunderstandings.
- Second, I do not agree very much with the form or distribution of point 5.2, line 591. I understand the authors want to focus on the most modern systemic treatment, but both ICI and TKIs are still chemical therapy and therefore chemotherapy. I consider it would be more useful in a review to treat the concept of systemic therapy in general, with a first section for immunotherapy, where to include both advanced and neoadjuvant treatment, a second section for EGFR inhibitors, as the main target therapy used in this type of neoplasia , and finally (and why not), a third section briefly considering what conventional (or anti-cell cycle) chemotherapy provides, as is still an option for some patients.
- Response: Dear colleague, thank you for your suggestion. The section 5.2., now called “Systemic treatment” have been reorganized, as suggested, and new information regarding conventional chemotherapy has also been added.
Reviewer 2 Report
Comments and Suggestions for Authors
This is an extensive attempt to summarize cutaneous squamous cell carcinoma (cSCC). There are a plenty of very basic statements such as how to prepare the surgical field when attempting to resect tumors of the scalp, clinical pictures of cSCC and HE stained pathology etc. The style and content rather qualifies for a textbook chapter than for a scientific paper. But even for a textbook chapter, this manuscript requires a thorough structure. For example, when and how regional disease should be screened in the stating process is not explained precisely.
The authors intended to focus on cSCC of the scalp but got lost as they included too many basic facts on general cSCC. In addition, by resuming controversial discussions on the safety margins, T-staging systems and others they definitively lost the common theme.
Comments on the Quality of English Languageneeds some corrections (e.g. prepositions, some wording mistakes)
Author Response
Dear colleague, thank you for your comments. Our intention was to summarize all the evidence regarding scalp cSCC found in the literature, and provide a global perspective of the state of the subject, not only addressed to dermatologists but also to oncologists, generalists and investigators, which are not always as familiarised with the topic. Besides, we wanted to highlight those areas where the evidence is scarce or lacking, in order to point out and guide further works that might be both interesting and useful in real clinical practice.
Event though, the introduction has been simplified, and some “general” facts have been removed, as well as “ Table 2”, where all 3 staging systems were listed.
Regarding “when and how regional disease should be screened”, this is described in Section 6.1. as stated: “…recommend imaging tests at diagnosis in patients without palpable lymphadenopathy on examination who present with high-risk cSCC, defined by a T score according to BWH equal to or greater than T2b or the presence of any of the risk factors proposed by the EADO”.
Besides, next paragraph discusses how both CT and ultrasound can be used for this purpose, depending on the suspicious of bone invasion or the presence of renal failure or other comorbidities. This information is also summarized in Algorithm 7.1.
Reviewer 3 Report
Comments and Suggestions for Authors
This review aims to evaluate the role of imaging and the therapeutic approaches, especially surgical treatment on scalp SCC. It is necessary and has the meaningfulness for the management of scalp SCC. However, some major and minor revision as follow should be made before publication.
1. The emphasis of this review is scalp SCC, therefore the part of introduction should be simplified. It is suggested that table 1 focuses on summarizing the clinical and histological risks on scalp SCC, not total cSCC.
2. Furthermore, table2 just lists the AJCC8, Salamanca and BWH Staging Systems for cSCC, lacking authors’ opinions or summaries, and is suggested to be removed.
3. It is suggested that the figures for clinical photos (figure 2), dermoscopy findings (figure 3) and histological findings (figure 4) should contain scalp SCCs with different stage including Tis (in situ), T1, T2 and T3, or low, moderate and high differentiation.
4. Surgical treatment is the important emphasis of this review. Authors review the surgery pattern, excision and histological margin assessment, Mohs surgery, reconstructive approaches on the scalp, et al. However, it is suggested to summary these contents into one table that is convenient for reader, eg. listing the recurrence rate with different excision scope and depth.
5. In section 6, it is suggested to supplement the review of dermoscopy on scalp SCC, especially for early SCC.
6. Others: Please note spelling mistakes, eg. line 42, revision? line 221,dermatoscopy?Line 355, section 5, imaging approach, should be section 6?
Comments on the Quality of English LanguageMinor editing of English language required
Author Response
Dear colleague, thank you for your comments.
- The emphasis of this review is scalp SCC, therefore the part of introduction should be simplified. It is suggested that table 1 focuses on summarizing the clinical and histological risks on scalp SCC, not total cSCC.
- Response: The introduction has been simplified. “Table 1” about risk factors, now is in the “scalp section” (section 2), and refers only to Risk factors of scalp cSCC. Information regarding histological margins in scalp cSCC has also been included in this Table.
- Furthermore, table2 just lists the AJCC8, Salamanca and BWH Staging Systems for cSCC, lacking authors’ opinions or summaries, and is suggested to be removed.
- Response: table 2 has been removed.
- It is suggested that the figures for clinical photos (figure 2), dermoscopy findings (figure 3) and histological findings (figure 4) should contain scalp SCCs with different stage including Tis (in situ), T1, T2 and T3, or low, moderate and high differentiation.
- Response: clinical photos and their footnotes have been revised and adjusted to show low-, moderate- and well differentiated scalp cSCC. Nonetheless, new dermoscopy and histological photos were finally not included due to length issues, and under the suggestion of reviewer 2, who recommends reducing the article’s length and focus on scalp cSCC.
- Surgical treatment is the important emphasis of this review. Authors review the surgery pattern, excision and histological margin assessment, Mohs surgery, reconstructive approaches on the scalp, et al. However, it is suggested to summary these contents into one table that is convenient for reader, eg. listing the recurrence rate with different excision scope and depth.
- Response: A new table (“Table 2”, in section 3. Surgical treatment) has been added to summarize the evidence of all different modalities in surgical treatment of scalp cSCC. The evidence regarding histological margins has been included in Table 1, the one focused on risk factors of scalp cSCC.
- In section 6, it is suggested to supplement the review of dermoscopy on scalp SCC, especially for early SCC.
- Response: in section 2.3. More information regarding dermoscopy, specially for early cSCC, has been added.
- Others: Please note spelling mistakes, eg. line 42, revision? Line 221,dermatoscopy?Line 355, section 5, imaging approach, should be section 6?
- Response: spelling mistakes were revised and corrected. Thank you.
Reviewer 4 Report
Comments and Suggestions for Authors
Comments: In this work, the authors outline the management of patients with scalp cutaneous squamous cell carcinoma (cSCC), including epidemiology, clinical presentation and diagnosis, imaging techniques, surgical and radiation treatments, systemic therapy for advanced cases and follow-up. The current guidelines lack specific recommendations for scalp cSCC, leading to inconsistent decision-making in multidisciplinary committees, especially for tumors with high risk factors or close margins. The article offers a thorough exploration of cSCC management, combining scientific rigor with practical considerations. The clarity, evidence-based approach, and focus on emerging therapies make it a valuable resource for professionals in dermatology and oncology. This review is well-organized, presenting information in a clear and systematic manner. The details about different surgical and non-surgical approaches are explained thoroughly, making it accessible to both medical professionals and researchers.
Author Response
Dear colleague, thank you for your kind remarks and comments.
Reviewer 5 Report
Comments and Suggestions for Authors
Dear Authors,
the manuscript provides a comprehensive and clear overview on the management of cutaneous squamous cell carcinomas of the scalp. The reading is enjoyable and fluent. Tables are appropriate and clear. Moreover, I find the proposed algorithms useful and reasonable. I enjoyed reading the paper.
I would recommend to add data on pembrolizumab (date of approval) and information by the regulatory agencies (FDA and EMA)
The paper provides a comprehensive overview on the management of cutaneous squamous cell carcinomas of the scalp, given the lack of specific recommendations for this specific location.
The topic is of interest for the scientific community in my opinion. Although there are other articles discussing about the management of cSCC, this article focuses on the location of the head that requires a more tailored approach based on the anatomical characteristics.
The tables are easily readable, and well- realized. Likewise, pictures are explicative. I found particularly useful the algorithms proposed by the authors.
Author Response
Dear colleague, thank you for your review and kind comments. We’ve just added some new information about Pembrolizumab, including date of approval.
Round 2
Reviewer 2 Report
Comments and Suggestions for Authors
many of my concerns are met
Comments on the Quality of English LanguageMS should be read through by a native English speaking expert.
Reviewer 3 Report
Comments and Suggestions for Authors
Authors have addressed most of my comments point by point.
Comments on the Quality of English LanguageMinor editing of English language required